# Visual attention towards food during unplanned purchases – A pilot study using mobile eye tracking technology

**Gerrit Hummel** [ID] **\*, Saskia Maier, Maren Baumgarten, Cora Eder, Patrick Thomas Strubich, Nanette Stroebele-Benschop**

Institute of Nutritional Medicine, Stuttgart, Germany

* gerrit.hummel@uni-hohenheim.de

## Abstract

This pilot study aims to investigate the relationships between consumers' weight status, energy density of food and visual attention towards food during unplanned purchase behavior in a real-world environment. After more than a decade of intensive experimental eye tracking research on food perception, this pilot study attempts to link experimental and field research in this area. Shopping trips of participants with different weight status were recorded with mobile eye tracking devices and their unplanned purchase behavior was identified and analyzed. Different eye movement measurements for initial orientation and maintained attention were analyzed. Differences in visual attention caused by energy density of food were found. There was a tendency across all participants to look at low energy density food longer and more often.

## 1. Introduction

This study investigates the relationships between an individuals' weight status measured as body mass index (BMI), energy density of food (ED) and visual attention (VA) towards food products during unplanned purchases in a real-life supermarket. The study adds to the body of literature that already investigated the relationships between food choice and VA towards food under experimental conditions [e.g. 1–8]. Furthermore, it extends this perspective by embedding it in the empirical approach of unplanned purchase behavior [9, 10]. There are several reasons that support investigating unplanned purchases as an appropriate way to explore VA towards food at the point of sale (POS). First, individual purchase decisions are in 40 to 70% of all purchases unplanned [11–13]. Second, researching unplanned purchase behavior means researching real food choices made at the POS in everyday life without any constraint. Thereby a realistic combination of bottom up and top down control of VA [14, 15], linked to different stages of cognitive effort during food choice behavior [16] takes place. By measuring eye movements, it is possible to obtain insight into participants' attention and infer linked cognitive processes [14, 17]. Third, exclusive researching of unplanned purchase behavior compared to all purchases, limits the number of relevant purchase decisions to a manageable amount for data preparation and analysis.

**Data Availability Statement:** The data file is available from the figshare database: Hummel, Gerrit (2020): et_super_long. figshare. Dataset. https://doi.org/10.6084/m9.figshare.12962912

**Funding:** The author(s) received no specific funding for this work.

**Competing interests:** The authors have declared that no competing interests exist.

Besides marketing strategies at the POS [11, 16], the individuals' physiological and psychological characteristics are also known as a driver of purchase and consumer behavior [18, 19]. Based on findings from former experimental studies, BMI [1, 2, 5, 8] and ED [20, 21] are considered as crucial drivers of VA towards food. However, findings regarding the relationship of BMI and **initial orientation** are mixed and not consistent. Castellanos et al. [1] found that participants with normal weight initially looked more often towards food than nonfood stimuli, whereas Werthmann et al. [8] found the opposite. Gearhardt et al. [2] showed a decreased initial attention towards fried food for participants with higher BMI compared to participants of lower BMI. With regard to **maintained attention**, Castellanos et al. [1] found that, especially in fed condition, participants with obesity maintained their increased VA towards food stimuli whereas the group with normal weight showed reduced VA in fed condition. The authors described this finding as "system reward dysregulation [. . .] that is manifested as altered attentional salience" [1, p. 1070]. In 2019, Segovia, Palma, & Nayga [22] studied the effect of food anticipation on cognitive function. Thereby they used eye tracking technology to examine how an anticipatory food reward affects VA. They found that participants with overweight or obesity spent more time looking at regular snacks compared to normal weight individuals in a condition without anticipatory effect. In contrary, other studies [2, 3] did not find differences in maintained attention towards different food cues. Recently, Liu, Roefs, Werthmann, and Nederkoorn [23] reanalyzed data from three studies using trial level bias scores. They found that participants with overweight or obesity showed larger variability in their attention compared to participants with normal weight. Overall, there seem to be differences in VA to food between people with different BMI. Consequently, we derive our first hypothesis for unplanned purchases from this assumption. *H1*: *Unplanned purchases that are made by participants with a lower BMI differ from purchases that are made by participants with higher BMI regarding their VA towards food.*

ED can be seen as another important influence for VA towards foods. Nijs et al. [5] revealed an attentional bias towards food pictures compared to nonfood stimuli across all participants with no significant difference between weight status groups. Freijy et al. [21] examined interaction effects between type of stimuli and energy density of the presented food. The authors interpreted the bias towards high calorie food pictures and away from high calorie words as a result of the differences in cognitive processing regarding words compared to pictures. Doolan et al. [20] revealed an attention bias for both gaze direction and duration towards high calorie (HC) foods compared to low calorie (LC) foods across all participants regardless of their BMI. A study by Hummel et al. [4] found no differences in attention between foods with high and low energy density but between food preparation types of low energy density foods. Wang et al. [6] were able to show that lean participants directed their gaze longer towards high sugar foods than low sugar foods whereas overweight participants showed no such bias. Thus, most studies showed biases in VA affected by different stimuli types or varying energy density. Accordingly, we propose *H2*: *There are differences in VA between unplanned purchases towards HC and LC food.* Taking the assumed bivariate associations together, we postulate *H3*: *There is an interaction effect between BMI of the participant and ED of food in regard to VA.*

It is likely that the different paradigms (free viewing, visual probe tasks, search tasks) used across former studies may have led to the divergent findings [4, 24]. While most studies were conducted under experimental conditions, the current study is one of only a few [25, 26] that use an innovative approach in a real-world environment to investigate the relationships between VA towards food, BMI and ED of food under realistic conditions.

## 2. Materials and methods

The study was conducted in accordance with the Declaration of Helsinki and approved by the ethical committee of the University of Hohenheim. Permission from the supermarket owner was also obtained.

### 2.1 Sample and participant recruitment

Store familiarity influences in-store navigation [27] and unplanned purchases [9, 10]. Thus, data collection and participant recruitment were carried out in one full service supermarket located on the outskirts of a large city in south Germany that consists of a floor area of 1350 m$^2$ with a comprehensive assortment of about 20.000 to 25.000 products. Since all participants had already visited the supermarket for at least one time before recruitment, it was assumed that the store was not totally unknown to them.

During recruitment, the participants were informed about the research procedure and after meeting inclusion criteria and signing written informed consent, participants could take part in the study. Participation was only permitted for persons between 20 and 65 years of age, who lived in a 2 to 5-person household, who usually went on one shopping trip per week (except for bread and small items), who followed no special diet, who had no food intolerance or other nutrition-related disorder and who did not wear glasses or contact lenses. All data was gathered from January to May 2017.

Each participant received a 50 € voucher for the supermarket after study completion. Twenty participants were recruited. Since one participant did not follow the instructions, data of this participant were excluded from further analysis.

### 2.2 Procedures

Participants were asked to document their individual shopping and purchase behavior by writing a weekly shopping list, by making only one major shopping trip per week (except for necessities such as bread or milk) and by collecting the grocery receipts for four weeks. Besides a documentation of each weekly shopping trip (with questions such as: "Who joined the shopping trip?" "When did the shopping trip start and end?"), all participants had to wear a mobile eye tracking device during two supermarket shopping trips. One of the trips was at the beginning of week one and the other one at the end of week 4 of the study. By recording the shopping trips with mobile eye tracking technology, no standardized purchase situations were generated, and no purchase was forced. The participants knew they would wear mobile eye tracking technology during the shopping trips and were instructed to conduct their shopping trips as usual.

### 2.3 Eye tracking apparatus and measurements

Tobii Pro Glasses 2 (Tobii AB, Sweden) recorded the purchase behavior (e.g. adding an item to the shopping cart) and eye movements of the participants at 50 Hz with an average measurement error of 0.6˚ - 1.2˚ of visual angle. Every recording of a shopping trip started after a calibration and individual adjustments. Maintained attention towards food was measured as fixation duration in seconds (s), visit duration in seconds (s), fixation counts and visit counts. These measurements can be seen as indicator for related information processing [17]. Furthermore, time to first fixation (in s) was measured as an indicator for initial orientation which shows automatic or unconscious responses regarding food [16]. Besides these eye movement measurements, self-reported measures, e.g. age, sex, family structures, income, height, weight and psychological scales such as the Big Five personality traits [28], Zimbardos Time

Perspective Inventory [29] and a German version of the short form from the self-control scale [30] were recorded in two online questionnaires. Normally, it is advisable not to rely on self-reported measurements for size and weight but to measure them. In this study, the contact to the participant took place only in the supermarket. Unfortunately, there was no protected area there that would have allowed enough privacy for appropriate and respectful body measurements of the participants. Accordingly, we decided to request the body measurements in online surveys.

The duration as well as the sequence of the different steps of the study and the intervals between them were standardized for all participants (see Fig 1). The intervals between shopping trips and surveys were chosen as long as possible to minimize mutual influences.

## 2.4 Data preparation and analysis

In our study, we have chosen a simple and thus quite reliable definition and measurement for unplanned purchases. All purchases that were not previously planned by writing them down on the shopping list were defined as unplanned purchases. After cross checking grocery receipts and shopping lists to identify all unplanned purchases, the unplanned purchases had to be mapped in the video material to generate computable data. Therefore, two coders were trained until reaching 99% coders agreement and a relative error variance of less than 0.4%, which can be classified as very high and sufficient reliability [4, 31].

To make data comparable, standardized product grids of every single purchase situation had to be generated. Within these grids, products and labels (that show additional information such as organic or gluten-free and price tags that were attached to the shelf) are arranged around the chosen product which is in the middle of the map (see Fig 2). As a result, data are comparable across all unplanned purchases with the limitation that certain information of the products (such as size, shape and color) are not available for further analysis. However, areas of interest (AOI) were marked without overlapping the edge of the represented products. In one last step, the entire data had to be mapped in Tobii Pro Lab (Version 1.76.9338). For the mapping procedure, fixations were defined via software implemented I-VT filter settings. Referring to former studies [4, 6, 32], fixations were defined as eye movements that showed a

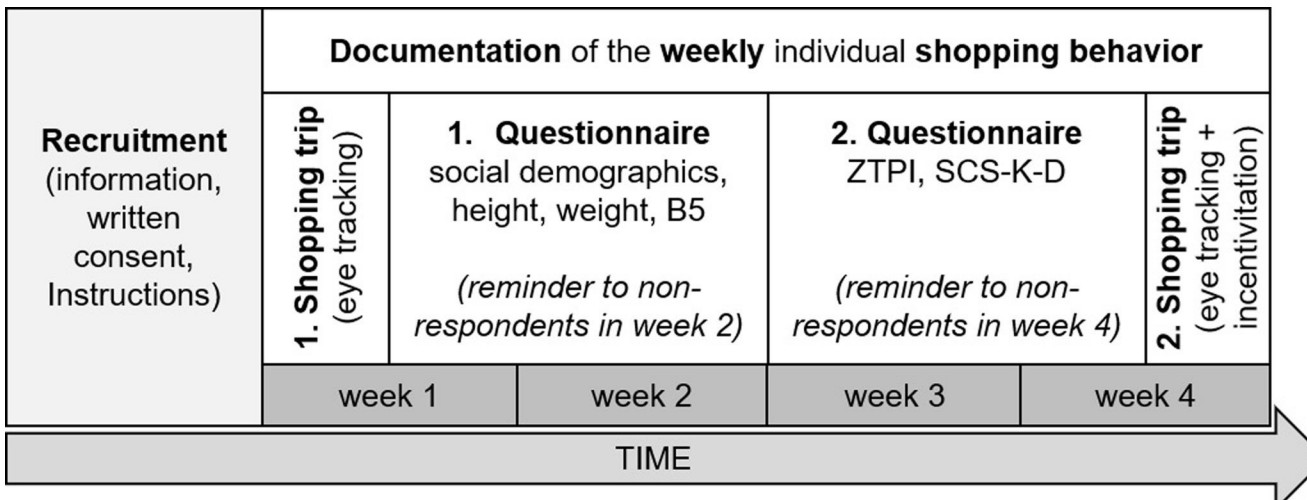

**Fig 1. Study procedure.** B5 = Big Five personality traits, ZTPI = Zimbardos Time Perspective Inventory, SCS-K-D = German version of the self control-scale.

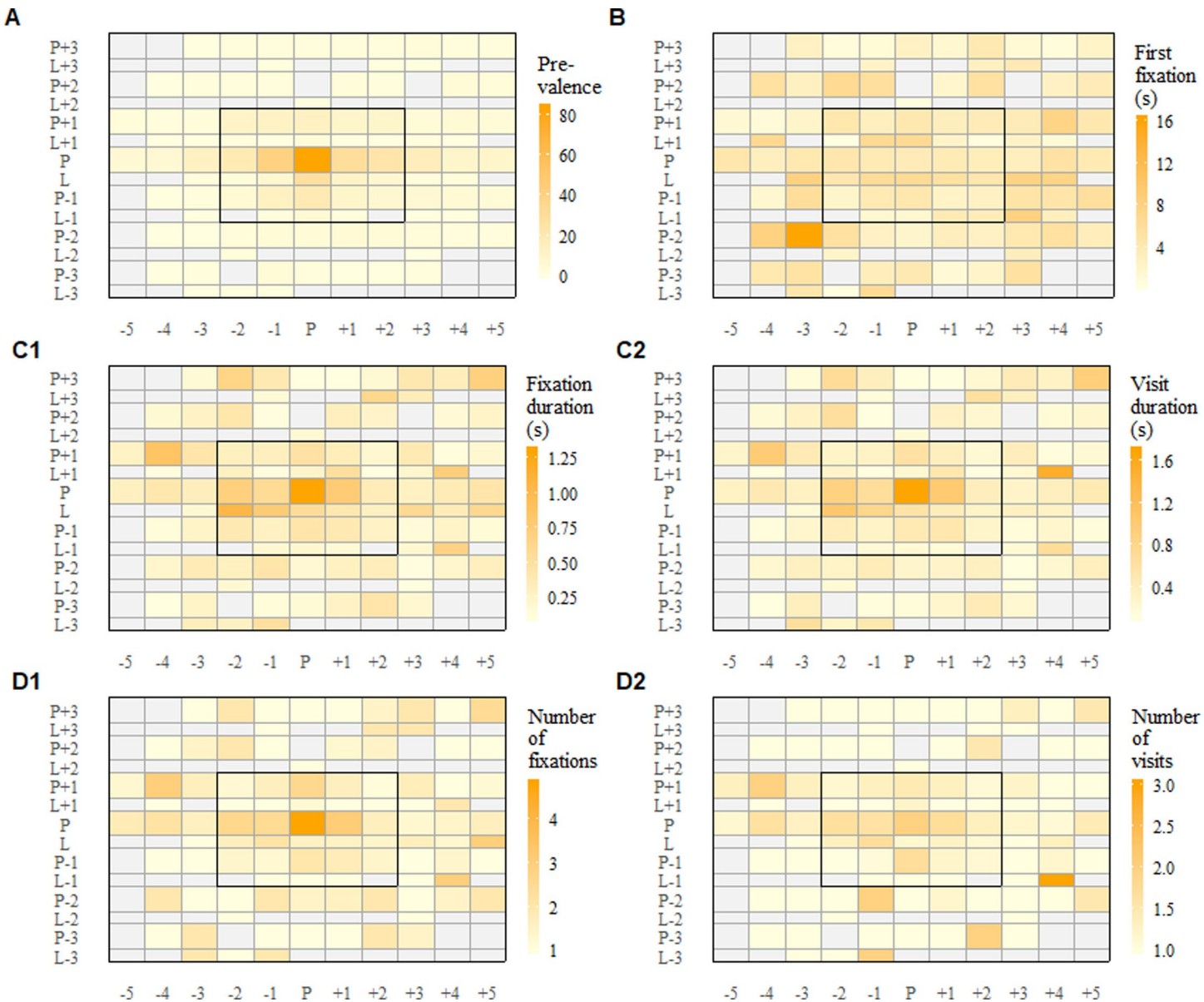

**Fig 2. Heat maps for visual attention towards products and shelf labels.** The bought product is placed in the middle of the map and can be identified by the letter P (for product) on the left side and the letter P (for product) below the heat map. The corresponding shelf label can be found directly under the bought product and can be identified by the letter L (label) on the left side and the letter P (product) below the heat map. The product one shelf higher and one row to the left of the bought product can be identified by the coordinates P+1 on the left side and -1 below the heat map. Heat maps show prevalence of attention across all purchases (A), time to first fixation measured in seconds (B), durations for fixations and visits measured in seconds (C) and number of fixations and visits (D). Darker colors in A, C and D indicate longer durations and higher number of fixations and visits. Darker colors in B indicate higher first fixations and delayed awareness. Grey fields mark areas without fixations.

velocity of less than 30˚/s, remained stable for at least 100 ms, and occurred during unplanned purchase behavior.

For further analysis, eye movement measurements were aggregated by building the means. Some single observations showed very high values regarding VA and duration of the unplanned purchase behavior. Therefore, data were adjusted for outliers that were outside the double standard deviation for total visit duration, fixation duration, duration of the unplanned purchase and total time during unplanned purchase behavior. After adjusting data for outliers,

a total number of 88 unplanned purchases were used for analysis. At first, further potential influences (such as pre-hunger ratings, duration of the shopping trip and relative shelf height of the product) were controlled for. Second, heat maps were visualized to investigate patterns of VA and the role of the bought product in comparison to other products and shelf labels. Third, independent-samples t-Tests (two-tailed) and Pearson's Chi-squared test with Yates' continuity correction were conducted to show differences between BMI and ED groups. Finally, linear mixed effect analysis was performed for the eye tracking parameter using the lmer function in the R package lme4 [33]. As fixed effects BMI (exact), energy density (exact), shopping companion (with partner) and sex entered the analysis. As random effect we had intercepts for subjects. Assumptions have been checked and they were satisfied. P-values were calculated by likelihood ratio tests of the full model against the model without the inspected effect. For time to first fixation, the inclusion of the random intercepts for subjects was leading to worse model. Also, multiple linear regression models had no acceptable fit. Steps one and three of the analytical plan were pre-specified to test the hypotheses. In addition, step two was added after building the heat maps to show that participants' attention was mainly focused on the chosen product and was not driven by additional information at the POS. Mixed effect models were calculated at the end of the analyses to consider possible influences of the repeated measurement and to control for the strength of the individual effects. All effects are reported as significant at $p < 0.05$. All analyses were conducted on the level of unplanned purchases using R statistics version 3.6 [34].

## 3. Results

### 3.1 Sample

Results were generated from 19 participants, wearing the eye tracking device during two different shopping trips. During these trips, 16 participants showed unplanned purchase behavior and three participants showed no unplanned purchase behavior (see Table 1). For further analysis, participants were classified as participants with normal weight (18.5 < BMI < 25, calculated as kg/m$^2$) and overweight or obesity (BMI $\geq$ 25). There were no participants with underweight (BMI < 18.5) in the sample. According to the sample mean (M = 259.33, SD = 227.33), food was divided into lower (< 260kcal/100g) and higher energy density food ($\geq$ 260kcal/100g). Participants with normal weight made 58 unplanned purchases (66%), while participants with a higher BMI made 30 unplanned purchases (34%). In 51 unplanned purchase choices (58%), the energy density of food was LC, and in 37 choices (42%), the energy density of the product was HC.

### 3.2 Control for additional influences

Food choice behavior is very complex [15]. It consists of a highly diverse mix of internal influences, aspects at the POS and external factors. Time spent on a shopping trip [19], the number of aisles that have been shopped [9, 13], the position of the bought product [17] hunger [1] and the number of accompanying persons during each shopping trip [35, 36] were examined before the main analyses were conducted. In a study over several weeks, time influences, learning effects or negative motivational reasons cannot be excluded. Therefore, we have examined whether a week effect can be observed. Results of these analyses are reported in Table 2. Considering participants' weight status, a significant difference was found for the accompanying person, $X^2 = 14.6$, p = < .001, Cramer's V = 0.4. In the group of participants with normal weight, the partner was the companion in only 2% of purchasing situations. In contrary, the partner was present in 43% of all purchasing situation in the group of participants with overweight or obesity. No other significant differences for any of the tested variables were found.

**Table 1. Sample and weight status groups' characteristics for unplanned purchase behavior and social demographics.**

|  | weight status groups | | | | sample | |
|  | normal weight | | overweight/ obese | | | |
|  | absolute | in % | absolute | in % | absolute | in % |
| --- | --- | --- | --- | --- | --- | --- |
| **number of participants** | 10 | 53 | 9 | 47 | 19 | 100 |
| . . . with unplanned purchase behavior | 9 | 56 | 7 | 44 | 16 | 100 |
| **number of unplanned purchases** | 58 | 66 | 30 | 34 | 88 | 100 |
| . . .of LC | 39 | 76 | 12 | 24 | 51 | 100 |
| . . .of HC | 19 | 51 | 18 | 49 | 37 | 100 |
| *social demographics* | M | SD | M | SD | M | SD |
| *age* | 35.1 | 10.0 | 34.6 | 10.7 | 34.8 | 10.0 |
| *gender* | absolute | in % | absolute | in % | absolute | in % |
| female | 8 | 80 | 5 | 56 | 13 | 100 |
| male | 2 | 20 | 4 | 44 | 6 | 100 |
| ***household*** | M | SD | M | SD | M | SD |
| household size | 2.4 | 1.2 | 2.4 | 0.5 | 2.4 | 0.9 |
| Income (per month and household) | absolute | in % | absolute | in % | absolute | in % |
| less than 1000,- € | 1 | 10 | 2 | 22 | 3 | 16 |
| 1001,- € - 2000,- € | 1 | 10 | 1 | 11 | 2 | 11 |
| 2001,- € - 3000,- € | 4 | 40 | 2 | 22 | 6 | 32 |
| 3001,- € - 4000,- € | 3 | 30 | 4 | 44 | 7 | 37 |
| more than 4000,- € | 1 | 10 | - | 0 | 1 | 5 |

**Note:** HC = high calorie foods, LC = low calorie foods.

Unplanned purchases of food with lower energy density occurred on average eight minutes earlier (M = 11.1 min, SD = 9.0) than unplanned purchases of food with higher energy density (M = 19.2 min, SD = 9.1), t(82) = -4.18, p < .001. Results showed also significant differences for relative time of purchase, described as the relation between the time of purchase and the shopping trip duration. On average, unplanned purchases of food with lower energy density occurred after 47% of the shopping time, while unplanned purchases of higher energy density food occurred on average after 71% of the shopping time, t(82) = -4.04, p < .001. There were no significant differences between unplanned purchases with products of lower and higher energy density for shelf position, how long it took to pick out the unplanned food choice or participants' self-rated hunger before shopping.

### 3.3 Heat maps and patterns of visual attention

The heat maps (Fig 2) show the diversity and patterns of VA towards different products and shelf labels in the product grid.

The prevalence of attention indicates the number of products that have been fixated for at least once. Fig 2 plot A shows a very strong central tendency towards the selected product. In 81 of 88 spontaneous purchases (92%), the bought product was at least fixated one time. There was also a difference in visual attention between products and labels. While information labels and price tags were on average focused three times during spontaneous purchase decisions (M = 3.1, SD = 5.1), the products themselves were focused more than twice as often (M = 7.3, SD = 11.9). There were also significant differences comparing products and labels across all purchases regarding fixation duration, t(45) = 3.12, p < .01 (see Fig 2, plot C1). While participants fixated products on average for 0.6 seconds (SD = 0.5), they fixated shelf labels for only

**Table 2. Selected potentially influencing variables on unplanned purchase behavior across weight status groups and energy density of the chosen foods.**

| | weight status (BMI) | | |
|---|---|---|---|
| | **normal weight (n = 58)** | **overweight/ obesity (n = 30)** | |
| **potential influences** | M (SD) | M (SD) | **t(p)** |
| time of unplanned purchase (in min) | 14.2 (8.9) | 15.1 (11.6) | -0.40 (0.694) |
| relative time of purchase | 0.6 (0.3) | 0.6 (0.3) | -0.85 (0.396) |
| unplanned purchase duration (in s) | 10.1 (6.3) | 10.9 (6.9) | -0.54 (0.590) |
| relative shelf height | 0.6 (0.3) | 0.6 (0.3) | 0.38 (0.709) |
| duration shopping trip (in min) | 26.2 (9.3) | 22.2 (9.3) | 1.91 (0.061) |
| pre hunger (10 cm VAS) | 3.1 (2.2) | 3.0 (2.3) | 0.11 (0.913) |
| | **n (in %)** | **n (in %)** | **$X^2$ (p)** |
| | | | **Cramer's V/φ** |
| with shopping companion | 4 (7) | 13 (43) | 14.6 (< .001) |
| | | | 0.4 |
| *accompanied by a child* | 4(7) | 1(3) | 0.0 (0.843) |
| | | | 0.0 |
| *accompanied by a partner* | 1 (2) | 13 (43) | 22.6 (< .001) |
| | | | 0.5 |
| number of purchases made during shopping trip one | 28 (48) | 9 (30) | 2.0 (0.156) |
| | | | 0.2 |
| | energy density | | |
| | **LC** (n = 51) | **HC** (n = 37) | |
| **potential influences** | M (SD) | M (SD) | **t(p)** |
| time of unplanned purchase (in min) | 11.1 (9.0) | 19.2 (9.0) | -4.18 (<0.001) |
| relative time of purchase | 0.5 (0.3) | 0.7 (0.3) | -4.04 (<0.001) |
| unplanned purchase duration (in s) | 10.6 (6.3) | 10.2 (6.7) | 0.28 (0.780) |
| relative shelf height | 0.6 (0.3) | 0.6 (0.3) | 0.31 (0.756) |
| duration shopping trip (in min) | 23.2 (9.8) | 27.0 (8.4) | -1.95 (0.054) |
| pre hunger (10 cm VAS) | 3.4 (2.6) | 2.7 (2.1) | 1.36 (0.178) |
| | **n (in %)** | **n (in %)** | **$X^2$ (p)** |
| | | | **Cramer's V/φ** |
| with shopping companion | 12 (24) | 5 (14) | 0.8 (0.367) |
| | | | 0.1 |
| *accompanied by a child* | 3 (6) | 2 (5) | 0.0 (1.000) |
| | | | 0.0 |
| *accompanied by a partner* | 10 (20) | 4 (11) | 0.7 (0.413) |
| | | | 0.1 |
| number of purchases made during shopping trip one | 23 (45) | 14 (38) | 0.2 (0.644) |
| | | | 0.0 |

**Note:** Participants' weight status was classified in participants with normal weight (18.5 < BMI < 25) and participants with overweight or obesity (BMI ≥ 25). Energy density of food was divided into low (LC < 260kcal/100g) and high (HC ≥ 260kcal/100g) energy density. Relative shelf height resulted from the relation between the shelf on which the product stands and the maximum number of shelves in the cabinet. VAS = visual analogue scale. Independent samples t-Tests (two-tailed) and Pearson's Chi-squared test with Yates' continuity correction are reported.

0.4 seconds (SD = 0.3). The same can be shown for visit duration (Fig 1, plot C2). Participants viewed products (M = 0.8, SD = 0.8) significantly longer than labels (M = 0.4, SD = 0.3), t(45) = 3.61, p < .001). There were also statistically significant differences between products and labels regarding the number of fixations (t(45) = 3.98, p < .001) and visits (t(45) = 5.88, p <

.001). Both showed higher values for products than for labels. Products were fixated on average 1.4 times (SD = 0.4), while means for labels showed only 1.1 fixations (SD = 0.3). Also, initial orientation showed differences between products on the one hand and labels on the other hand. Products were fixated on average after 2.9 seconds (SD = 2.2) for the first time while labels were fixated much later (on average after 4.6 seconds, SD = 3.5).

### 3.4 Group comparisons for BMI and energy density

Results of the group comparisons for spontaneous purchases made by participants with overweight or obesity and normal weight and purchases of food with higher and lower energy density can be seen in Fig 3. Purchases made by participants with overweight or obesity took a longer time to first fixation (M = 5.4 s, SD = 4.9) than purchases that were made by participants with normal weight (M = 2.8 s, SD = 2.8), t(38) = -2.47, p < .05. In contrast, purchases made by participants with overweight or obesity had a lower visit duration (M = 1.3 s, SD = 0.8) than purchases made by participants with normal weight (M = 1.9 s, SD = 1.7), t(81) = 2.01, p < .05. Thus, participants with normal weight noticed the food of their choice earlier and looked longer at it than participants with overweight or obesity (see Fig 3). Therefore, the purchases of the two weight status groups differ in terms of initial and maintained attention and H1 can be confirmed.

LC foods were on average looked at for 1.9 seconds (SD = 1.8), while HC foods were looked at for only 1.3 seconds (SD = 0.8) before they were chosen, t(71) = 2.03, p < .05. All participants showed more fixations (M = 5.5, SD = 5.0) towards LC food than towards HC food (M = 3.8, SD = 2.5), t(72) = 1.98, p = 0.05. Participants showed also more visits towards LC food (M = 2.2, SD = 1.3) than towards HC food (M = 1.7, SD = 0.7), t(74) = 2.16, p < .05. Overall group comparisons show that products with a lower energy density were fixated longer and more often than products with higher energy density.

### 3.5 Regression analysis to estimate maintained attention

BMI and energy density were included into the mixed model regression as fixed effects. Since gender is not equally distributed in our sample (see Table 1) and gender also had an influence on VA in other experimental studies [4, 32], gender was included in the model as a fixed effect. The presence of a shopping companion might also have a significant influence for VA (see Table 2). For regression analyses we must be aware of repeated measures with multiple responses from the same subject. Fig 4 shows the individual variation for the measurements of maintained VA (durations and counts for fixations and visits). The figure shows different participants having slightly different eye movements. For example, participant 18bT27 shows an average fixation duration of about 2 seconds while 18bT16 has an average fixation duration of less than one second (see Fig 4). Due to this issue the assumption of independence is violated. By using mixed models, the individual differences can be considered as random intercepts for each participant. And therefore, using mixed models can fix issues of non-independence.

For the last analyses, mixed models for measures of maintained attention (durations and counts) were calculated. Results for durations and counts are shown in Table 3. Unfortunately, the inclusion of the random intercepts for subjects is leading to a worse model that did not converge for time to first fixation. Also, using a multiple linear regression did not work since the overall F-Test was not significant in all tested models (even without sex and shopping companion).

The overall model predicting fixation duration successfully converged. The model's intercept is at 3.09 seconds (SE = 1.37, 95% CI [0.40, 2.69], $X^2(1) = 4.58$, p<0.05). Within the model, the effect of BMI and gender are not significant (p>.05). The effect of energy density is

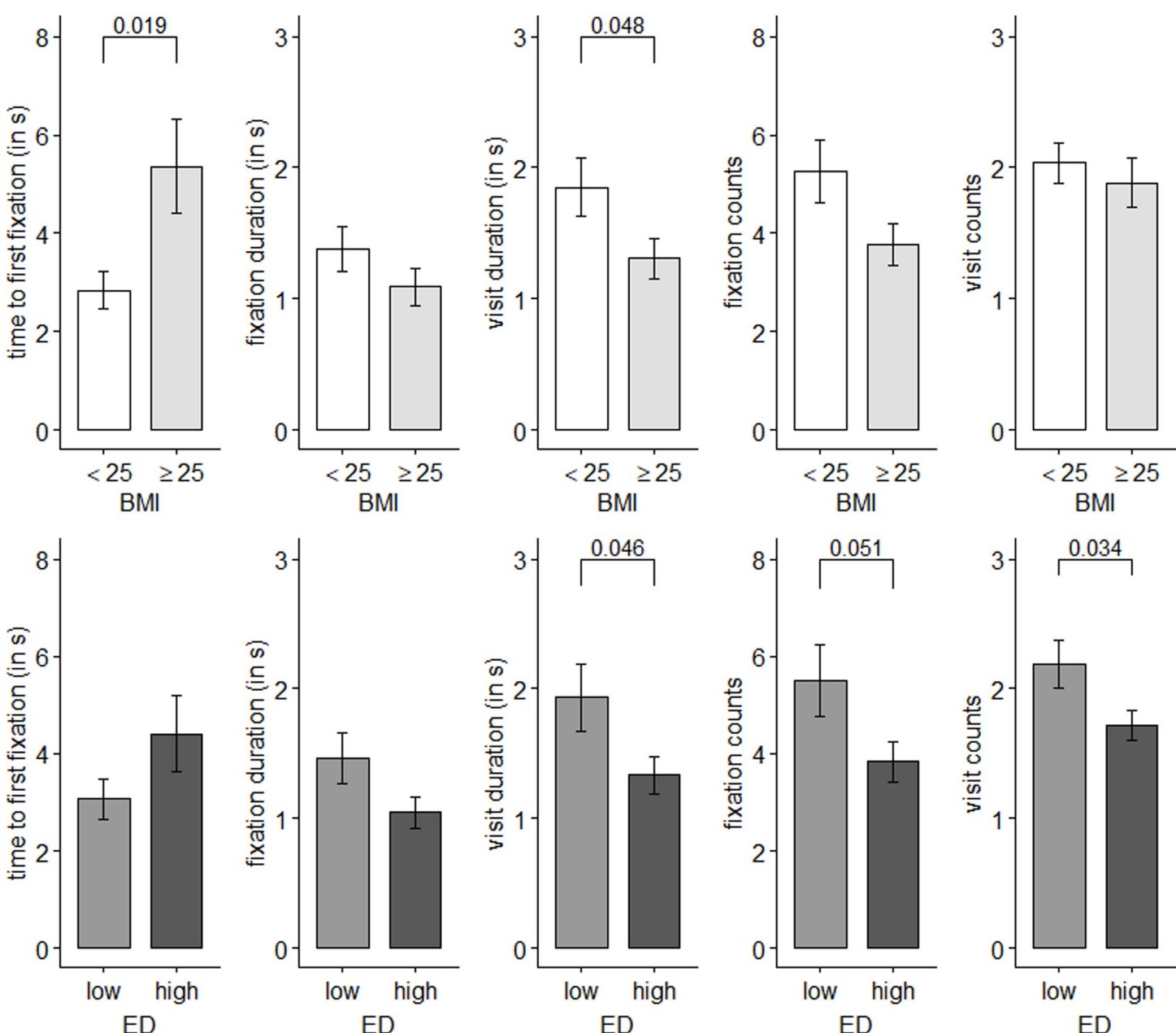

**Fig 3. Visual attention towards the chosen product during unplanned purchase behavior.** Black bar plots show means and standard errors for unplanned purchases (n = 81) that have been made by participants with normal weight (18.5 < BMI < 25) and overweight or obesity (BMI ≥ 25) and for low calorie food (< 260kcal/100g) and high calorie food (≥ 260kcal/100g). ED = energy density. Independent-samples t-Tests (two-tailed): p-values above the brackets show differences between groups.

significant (estimate = -0.001, SE = 0.001, 95% CI [-0.002, -0.000], $X^2(1)$ = 5.39, p<0.05) and can be considered very small. The effect of the presence of a companion is significant (estimate = -1.21, SE = 0.54, 95% CI [-2.27, -0.14], $X^2(1)$ = 4.58, p<0.05) and can be considered small.

The overall model predicting visit duration successfully converged. The model's intercept is at 3.67 seconds (SE = 1.53, 95% CI [0.68, 6.67], $X^2(1)$ = 5.32, p<0.05). Within the model, the effect of BMI, the presence of a companion and gender are not significant (p>.05). Only the effect of energy density is also significant for visit duration (estimate = -0.001, SE = 0.001, 95% CI [-0.002, -0.000], $X^2(1)$ = 4.19, p<0.05) and can be considered very small.

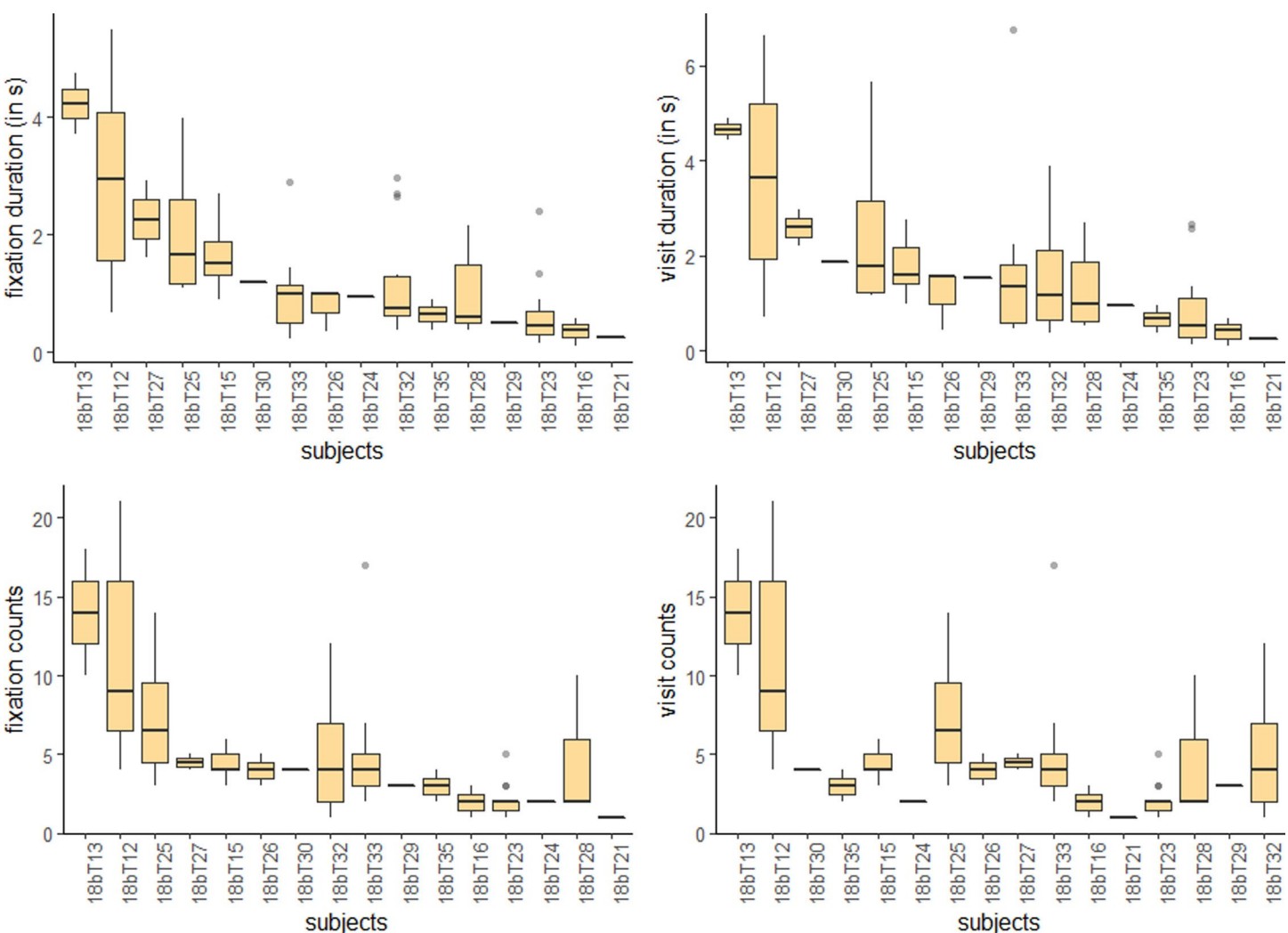

**Fig 4. Individual differences for eye tracking measurements.** Boxplots show individual measures for maintained attention (n = 83).

The overall model predicting fixation count successfully converged. The model's intercept is at 10.19 counts (SE = 4.74, 95% CI [0.89, 19.49], $X^2(1) = 4.16$, p<0.05). Within the model, the effect of BMI, the presence of a companion and gender are not significant (p>.05). The effect of energy density for fixation counts is significant (estimate = -0.004, SE = 0.002, 95% CI [-0.008, -0.003], $X^2(1) = 4.19$, p<0.05) and can be considered very small.

The overall model predicting visit count successfully converged. The model's intercept is at 3.66 counts (SE = 1.16, 95% CI [1.40, 5.93], $X^2(1) = 8.40$, p<0.01). Within the model, no tested effect is significant (p>.05).

While the pairwise comparisons have pointed to influences of BMI and energy density, the complex regression models only show an influence of energy density on maintained attention. Consequently, H1 cannot be confirmed whereas H2 can be confirmed. Models with and without interaction effects for BMI and energy density were calculated. In none of the cases did the addition of an interaction effect lead to an improvement of the model. Therefore, it can be assumed that for our models no interaction effects between BMI and energy density exist. H3 can therefore not be confirmed.

**Table 3. Results of the linear mixed-effects models of maintained attention towards the bought product during unplanned purchases.**

| | fixation duration (s) | | | visit duration (s) | | |
|---|---|---|---|---|---|---|
| **fixed effects** | **estimate** | **SE** | **X² (p)** | **estimate** | **SE** | **X² (p)** |
| intercept | 3.09 | 1.37 | 4.58 (0.032) | 3.67 | 0.63 | 5.32 (0.021) |
| BMI (exact) | -0.02 | 0.05 | 0.19 (0.658) | -0.03 | 0.06 | 0.25 (0.620) |
| energy density (exact)[t] | -0.001 | 0.001 | 5.39 (0.020) | -0.001 | 0.001 | 4.19 (0.041) |
| companion (partner) | -1.21 | 0.54 | 4.58 (0.032) | -1.21 | 0.62 | 3.65 (0.056) |
| gender | -0.76 | 0.52 | 2.01 (0.156) | -0.69 | 0.62 | 1.20 (0.272) |
| **random effect** | **variance** | **SD** | | **variance** | **SD** | |
| participant | 0.47 | 0.69 | - | 0.40 | 0.63 | - |
| residual | 0.71 | 0.85 | - | 1.60 | 1.27 | - |
| | **fixation counts** | | | **visit counts** | | |
| **fixed effects** | **estimate** | **SE** | **X² (p)** | **estimate** | **SE** | **X² (p)** |
| intercept | 10.19 | 4.74 | 4.16 (0.041) | 3.66 | 1.16 | 8.40 (0.004) |
| BMI (exact) | -0.11 | 0.19 | 0.33 (0.564) | -0.02 | 0.05 | 0.26 (0.611) |
| energy density (exact)[t] | -0.004 | 0.002 | 4.29 (0.038) | -0.001 | 0.001 | 2.49 (0.115) |
| companion (partner) | -2.94 | 1.90 | 2.33 (0.127) | -0.71 | 0.47 | 2.23 (0.135) |
| gender | -1.10 | 1.86 | 0.35 (0.556) | -0.76 | 0.47 | 2.43 (0.119) |
| **random effect** | **variance** | **SD** | | **variance** | **SD** | |
| participant | 4.72 | 2.17 | - | 0.24 | 0.49 | - |
| residual | 11.78 | 3.43 | - | 0.87 | 0.93 | - |

**Note:** For each eye tracking variable a separate mixed effect model has been conducted. Gender: 0 = male, 1 = female. Presence of a companion: 0 = none, 1 = partner. P-values were calculated by likelihood ratio test of the full model against the model without the effect that has been inspected. Each p-value was calculated as likelihood ratio test of the full model against the model without the according effect. P-values are not given for covariance parameters. [t]Since estimates and standard errors for energy density are very small they are rounded to three decimal places instead of two.

## 4. Discussion

This pilot study is the first to examine the association of BMI, ED and VA during unplanned purchases in a real-world setting. This innovative approach is characterized by the examination of unforced shopping behavior in a real supermarket. Therefore, findings from this study can confirm and enhance knowledge of former experiments regarding the relationship between VA and foods.

The first analyses revealed differences regarding the bought products of lower energy density. Participants with normal weight made 76% of the unplanned purchases of lower energy density foods, while only 24% of the unplanned purchases of lower energy density foods were bought by participants with overweight or obesity. In contrast, there was no difference between both groups regarding food with higher energy density. Referring to a study by Kaisari et al. [37], these findings might imply a healthier mindset for participants with lower BMI.

There were also differences in the time of the selected unplanned purchases between LC and HC food. On average, LC food was bought earlier during the shopping trip than HC food. This finding mirrors the general product arrangement in supermarkets. As in most supermarkets in Germany and elsewhere, fresh produce (LC food) are displayed in the first section when entering the store. In contrast, sweets, salty snacks such as potato chips, fast food and other HC foods can be found in the later sections.

In addition, we found that participants with overweight or obesity shopped more often with a partner than participants with normal weight. Already in 2005, Luo was able to show that the presence of peers or family members may influence (unplanned) purchase behavior.

The stronger the ties between the family members or peers, the stronger is the potential influence of the companions. In this study, we cannot exclude the possibility that the mere presence of companions has already influenced the perception or food choice. However, we have tried to exclude direct influences on shopping behavior. The participants were briefed not to let themselves be influenced by the companion while shopping. Furthermore, all spontaneous purchases where an influence could be triggered by visual or auditory stimuli of the accompanying person were excluded from the analyses (this could be the case if, for example, a partner points out that he wants something by telling about it or pointing to a product).

In the second stage of the analysis, heat maps revealed differences between VA towards shelf labels and products. Products received significantly more attention than information or price labels. Recently, some studies [38, 39] showed comparable results for labels on packages. Song et al. [38] showed that less than 50% of the participants in their study evaluated product information and a large number of participants did not even recognize them in a natural shopping environment. They concluded that most of the food choices were made based on previous experience and habits rather than information prompts on product labels. Besides the differences between VA towards products and labels, heat maps also revealed a strong central focus towards the bought product. Since the variety and heterogeneity of the information on the shelf labels was relatively high, it cannot be assumed that a difference in the density of information between shelf labels and products led to the differences in VA. Similar to findings from previous studies with experimental setups [40] and real-world settings [6, 25], heat maps from the current study confirmed that the duration and number of fixations and visits can be seen as a good predictor for consumers' choices.

In the last step of the analysis, VA towards the chosen product was estimated across the different weight status groups, energy density groups and a combination of both. Analysis for unplanned purchases made by participants with different weight status showed differences in the time to first fixation and visit duration. During unplanned purchases made by participants with lower BMI, the chosen product was fixated earlier and viewed longer than in purchases made by participants with higher BMI. Especially the earlier fixation of the chosen product by participants with lower BMI stands in contrast to findings from previous experimental studies [1, 3, 8]. One reason for these contrary findings might be the differences in the study setup. While the number of simultaneously presented stimuli in experimental settings is small ($2 < n < 5$), the number of products and alternatives in a real-world setting is immense. Therefore, the time to first fixation is inevitably higher in the current study within a natural environment than in experimental studies. Gidlöf et al. [26] specified a natural decision segmentation model that defines a longer orientation stage at the beginning of every decision-making process. Following their idea, initial orientation, as it was measured in the current study, cannot be interpreted as an unconscious inclination towards certain foods, but rather as a conscious search for foods and alternatives, that takes place within the first seconds of a food choice.

With regard to the ED of foods, differences in duration and number of visits were found. During LC purchases, the chosen products were viewed significantly longer and more often. There was a tendency to look longer towards LC products across the entire sample. Especially the combination of the participants' weight status and energy density of the product affected the fixation and visit duration towards food. Purchases made by participants with lower BMI had higher visit durations towards LC food and lower visit durations towards HC food, while purchases made by participants with higher BMI had higher visit durations towards HC foods and lower visit durations towards LC food. These findings appear to confirm the assumption that the mindset towards food affect VA in natural environments [37].

While the pairwise comparisons have pointed to influences of BMI and energy density, the complex regression models only showed an influence of energy density on maintained

attention. The influence of the energy density is very constant and can be observed for almost all measurements of maintained attention (except visit counts). Higher energy density led in almost all models to lower visit durations or less fixation counts. Regression model for fixation duration also suggests that the presence of a partner leads to lower fixation durations. The influence of the BMI disappears completely in the regression models and is also not found in any interaction effect.

## 5. Conclusions

To the authors' knowledge, these are the first results available that focused on the relation of unplanned purchase behavior, weight status and energy density of foods in a real-world setting. In addition to former findings from experimental studies, the results from the current study might indicate that especially energy density of food plays a key role in VA during unplanned purchases. But these findings must be interpreted with caution. Previously (see again Table 2), the relationship between the arrangement of high and low calorific food in the supermarket and the time of spontaneous purchases of these foods has already been pointed out. It would also be plausible that low-calorie foods were looked at longer and more frequently, since the participants were not yet under time pressure at this time of purchase. In the current study, the focus was on external validity. Accordingly, we refrained from changing factors that the natural setting dictates. Under experimental conditions, the starting point of each shopping trip could have been varied to experimentally control for the influence of time and time pressure. In this study, the study team decided against this option. Instead, the time of purchase was recorded, and it was statistically controlled whether a variation of this influence leads to differences in the eye movement measurements during the shopping trips. The results now provide evidence that, in addition to the relationship between energy density and eye movement measurements, there is also a relationship between the timing of spontaneous purchase and eye movement measurements. Which influence has the larger effect remains unclear at this point. Here we have reached the limits of an observational study. Nevertheless, this finding provides the opportunity for further experimental studies to investigate this phenomenon in more detail.

This pilot study has further fundamental limitations. Since the number of unplanned purchases was difficult to predict and the recruitment of the participants for the eye tracking study challenging, results are based on 88 unplanned purchases made by 16 individuals. Consequently, results based on this sample and analysis are highly preliminary and have to be confirmed by further studies with larger samples. Even the results of more complex calculations such as regression models can only be interpreted with extreme care due to the small number of cases. In each model, the number of chosen predictors is at the edge of overfitting the model.

Where the differences in attention and interest for different types of food come from cannot be answered by this study. Both habit formation as well as genetic factors might play a role [41, 42]. While genetic factors are difficult to change, positive eating habits can be formed from early age [43–45]. Therefore, the importance of looking into unplanned food purchasing behavior in relation to an individual's weight status is obvious. If attention and selection differences between weight status groups regarding food types can be found, shopping behavior and in particular the control of unplanned purchases should play a more dominant role in both research and treatment. One potential intervention tool could be health goal priming [46, 47] given its potential to support conscious forms of decision making as a method of unconscious regulation. Up to now, the relevance of food selection while shopping does not seem to be adequately mentioned or discussed in weight or obesity management [48, 49]. Further research in

this area is necessary and might increase opportunities for behavioral changes in both prevention and treatment.

## Acknowledgments

We want to thank the supermarket owner, who allowed us to use his supermarket for recruitment and data collection.

## Author Contributions

**Conceptualization:** Gerrit Hummel, Nanette Stroebele-Benschop.

**Formal analysis:** Gerrit Hummel.

**Investigation:** Saskia Maier, Maren Baumgarten, Cora Eder, Patrick Thomas Strubich.

**Methodology:** Gerrit Hummel.

**Project administration:** Saskia Maier, Maren Baumgarten, Cora Eder, Patrick Thomas Strubich.

**Supervision:** Gerrit Hummel, Nanette Stroebele-Benschop.

**Visualization:** Gerrit Hummel.

**Writing – original draft:** Gerrit Hummel, Nanette Stroebele-Benschop.

**Writing – review & editing:** Gerrit Hummel, Nanette Stroebele-Benschop.

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
