## [Decision Letter · Decision Letter 0]

27 Jul 2020

PONE-D-20-12685

Visual attention towards food during

unplanned purchases – A pilot study using mobile eye tracking technology

PLOS ONE

Dear Dr. Hummel,

Thank you for submitting your manuscript to PLOS ONE. After careful consideration, we feel that it has merit but does not fully meet PLOS ONE’s publication criteria as it currently stands. Therefore, we invite you to submit a revised version of the manuscript that addresses the points raised during the review process.

We look forward to receiving your revised manuscript.

Kind regards,

Zhifeng Gao

Academic Editor

PLOS ONE

Journal Requirements:

Reviewers' comments:

Reviewer's Responses to Questions

**Comments to the Author**

1. Is the manuscript technically sound, and do the data support the conclusions?

Reviewer #1: Partly

Reviewer #2: Partly

Reviewer #3: Yes

2. Has the statistical analysis been performed appropriately and rigorously? 

Reviewer #1: No

Reviewer #2: No

Reviewer #3: Yes

3. Have the authors made all data underlying the findings in their manuscript fully available?

Reviewer #1: No

Reviewer #2: No

Reviewer #3: No

4. Is the manuscript presented in an intelligible fashion and written in standard English?

Reviewer #1: Yes

Reviewer #2: Yes

Reviewer #3: Yes

5. Review Comments to the Author

Reviewer #1: Referee Report for PONE-D-20-12685

Visual attention towards food during unplanned purchases— A pilot study using mobile eye tracking technology

This papers uses a field experiment to examine the relationship between individuals’ weight status, food energy density, and visual attention during unplanned purchases. Results indicate that participants with higher BMI show an attention bias towards high-calorie foods compared to low-calorie foods. The opposite effect is found for subjects with lower BMI.

There is much to like about this paper. It addresses a very important public health problem (unhealthy eating) in a novel way. It considers distributional effects across relevant groups (i.e. BMI categories), which are generally understudied. Having said that, I have major concerns which are mainly related to the lack of information on the experimental procedures and the small number of observations. Please find my concerns below.

Main Comments

1. My biggest concern relates to sample size. There are two weight classifications (high and low BMI), with less than 10 subjects in each category. This is a low number unlikely to be supported by an ex ante power calculation. I understand that obtaining a large number of participants in a real setting, particularly subjects with high BMI, can be somewhat of a challenge. But any statistical analysis based on such few subjects would, at best, be highly preliminary. I would encourage the authors to increase their sample size significantly.

2. Subjects are classified into two weight categories based on self-reported measures of height and weight. Since BMI classification is a main component of this study, relying on self-reported measures is not ideal. If the authors are to increase sample size, I suggest they look into ways to collect weight and height measures during the experiment; for example, experimenters could collect these measures at the end of the last shopping trip, right before payment.

3. I find the description of experimental procedures to be exceptionally sparse. When was the eye tracking data collected? Was the same weeks used for all subjects? (e.g. eye tracking data was collected for all subjects in weeks 2 and 4), Did subjects knew they would be using eye tracking glasses during those trips beforehand?

4. How many online questionnaires were collected by subject and when were these implemented? What information did they collect besides socio-economic characteristics? This is important because the type of questions could prime subjects and affect their subsequent purchasing behavior. For example, it would make a big difference if the self-reported weight and height measures were collected at the beginning or at the end of the experiment as asking for subjects’ weight might prime them towards specific food products or product quantities.

5. Who are the participants? The authors mentioned that socio-economic characteristics were collected in the online survey (e.g. gender, age, income); however they do not provide any description of the sample population. I find it a significant omission for an experimental paper to not present a summary of the demographic profile of the sample. This is not necessarily a criticism of the experiment procedures, but rather the written summary of the procedures. I suggest the authors provide a table with a summary of the demographic profile for all the sample and by BMI category (normal weight and overweight/obese).

6. Each participant received a 50 € voucher as compensation fee at the end of the experiment. Was there another incentive throughout the experiment? If subjects knew they would be receiving the voucher for completing the study, there was no incentive for them to truthfully report their preferences during each shopping visit (i.e. while making shopping lists and purchases). The compensation should have been split into weekly payments.

7. Why was participation restricted to households with 2-5 members? Were participants shopping by themselves? This is important as the presence of a second individual might steer participants toward specific products. For example, Papoutsi et al. (2013) show that children’s pestering power strongly affects parents in making unhealthier food choices. I suggest the authors use data from the online questionnaires (e.g. Who joined the trip?) to control for the number of people in the shopping trips as well as household size.

8. Subjects were asked to write a weekly shopping list and collect grocery receipts for four weeks. When were the receipts and lists collected by the experimenter? If these were collected at the end of the experiment, subjects had the opportunity to revise their shopping lists as much as they wanted. Also, there is the possibility that after 1-2 weeks subjects got an idea of the purpose of the experiment and started adjusting their behavior. I suggest the authors control for week effects.

9. The results in Table 1 and Table 2 should be split by BMI and energy density. For example, I would like to see the number of LC and HC unplanned purchases for each BMI category, same for all statistics reported in Table 2. This might cause the significant effects in Table 2 to disappear due to the small sample size, which highlights the need to collect more data.

10. Throughout the paper, there is no mentioning of the type of statistical tests used (t-tests, one or two-sided, etc.) when statistical analysis is performed. This information needs to be clarified, particularly in Tables and Figures. The authors refer to p-values in Table 2 and Figure 2– what are the statistical tests? The authors should include the relevant missing information in the table notes and figure captions so that they are self-contained.

11. The analysis is generally lacking in statistical rigor. The results are primarily based on pairwise-comparison tests and I think there are several important factors that need to be controlled for using model specifications/regressions. For example, I would strongly encourage the authors to explore whether the results vary over time (by week); are the findings consistent if one looks at early purchases (e.g. first 2 weeks) vs. later purchases (2 latest). The authors should also control for time-of-the-day effects and individual characteristics such as hunger level, gender, household size, income, eating habits.

12. Related to my previous comment, the authors are using the observations (unplanned purchases) made by the same individual in different weeks as independent observations. I suggest they consider using models for panel data that cluster the standard errors at the individual level and control for all factors described in the previous comment. They can also test for interaction effects between weight status and energy density using these models rather than ANOVA tests (or complement both).

13. The authors compare attention bias towards the bought product and the labels. What information was provided in the shelf labels? If the only information provided was the product price, a possible explanation for the lower amount of time spent looking at the labels compared to the product could be the familiarity with this attribute. It is reasonable to think that since individuals compare product prices on a regular basis, they might not need as much time to process price information. I think the authors could make more use of the eye tracking data by creating AOIs for the product labels instead of shelf labels. They could explore whether subjects fixate longer on calorie content or health claims such as low-fat, sugar reduced, fat-free, etc. and relate this to unplanned food choices.

14. It would have been interesting to see more results related to heterogeneity of unplanned purchases/visual attention of individuals who were more/less hungry according to the scale the authors collected. There is a vast literature on how hunger might produce different effects, and this paper could have something to say about this.

Other comments

1. The relationship between visual attention and food choice across BMI categories is an important contribution that has been highly understudied. Only few studies that have examined such distributional effects (Segovia et al. 2019). This is worth stressing more in the paper.

2. The authors should be consistent in the terminology used for BMI categories, sometimes they refer to the groups as high vs. low BMI, sometimes as BMI<25 vs. BMI>25, and other times as normal weight vs. overweight/obese.

3. Are there underweight subjects in the sample? (BMI< 18.5). If so, they should not be part of the normal weight group.

4. Page 15, lines 357-358: sentence makes no sense. Please edit.

5. The limitations of the study can be discussed in the conclusions section, no need for a separate section.

References

Papoutsi, Georgia, Rodolfo Nayga, Panagiotis Lazaridis, and Andreas Drichoutis. "Nudging parental health behavior with and without children's pestering power: Fat tax, subsidy or both?." (2013).

Segovia, Michelle S., Marco A. Palma, and Rodolfo M. Nayga Jr. "The effect of food anticipation on cognitive function: An eye tracking study." PloS one 14, no. 10 (2019): e0223506.

Reviewer #2: This paper investigated the link between visual attention and unplanned food purchase decisions in a real-world environment, while incorporating conceptually relevant variables such as participants’ weight status and cues presented on food labels such as energy density. The authors adequately reviewed the literature and formed the hypotheses in a manner that help to shed light on earlier inconsistencies in the literature on this topic. An important contribution, as the authors also highlighted, is the use of visual attention data generated in a real-world environment as opposed to a controlled laboratory. On the flip side, experimenting in a real-world setting represents a host of challenges that can interfere with the investigation that aims to link visual attention to other individual-specific characteristics or product-specific attributes. Additionally, while the introduction of the paper is well prepared, there are some shortcomings in the description of the experimental procedures. Please see my review comments below.

Comments

- The experiment design and hence the findings suffer from a major issue due to the product arrangement in a supermarket. As indicated in Lines 313-315, if the LC foods are offered in the first section, while the HC foods are located in the later sections. This gives a basis to consider that consumers may face less time pressure or constraints in choosing LC foods (i.e., beginning of the shopping experience) and more time pressure/constraint in choosing HC food (i.e., toward the end of the shopping experience). Because of this arrangement, consumers may spend more time and pay more visual attention to LC foods, while paying less visual attention to HC foods. In other words, we don’t know for sure whether the differences in visual attention between HC and LC foods (and across BMI groups) were caused by the product arrangement or the differences in energy density that the authors tried to identify in the study. Generalizing statement such as “There was a tendency across all participants to look at low energy density food longer and more often.” (Lines 12-13) need to be interpreted with caution. The authors may want to acknowledge this as a major limitation of this study.

- Lines 283-287: The authors compared fixation duration between low BMI participants for LC, high BMI participants for LC, and low BMI participants for HC. From my point of view, the authors should conduct a complete between-group and within-group comparison to capture the interaction between BMI and energy density on all VA measures. Using fixation duration as an example, between-group comparison should be conducted for a) between low BMI participants’ fixation duration for LC foods, and high BMI participants’ fixation duration for LC foods, b) between low BMI participants’ fixation duration for HC and high BMI participants’ fixation duration for HC. Within group comparison should be conducted for a) between low BMI individuals’ fixation duration for LC and HC, b) between high BMI individuals’ fixation duration for LC and HC foods.

- The authors may consider adding a table reporting the demographic variables comparison between the low and high BMI groups as we know BMI is closely related to individual characteristics such as age and income. This information is completely missing in the current version. Demographic information is particularly important for this study because the sample size is very small.

Other comments or clarification needed

- Section 2.4. The authors identified consumers’ unplanned purchases by checking their grocery receipts for the past four weeks. As shoppers ourselves, we know perfectly that there are times when last-minute items pop up when we are already at the store, even with a shopping list. Since this is a major variable of interest and used to test two of the three hypotheses, I have concerns about the reliability of this procedure. Please comment.

- Section 3.1. The number of participants in this study is a concern. I am well aware of the difficulties associated with implementing eye tracking studies, especially at a grocery store. However, without reporting the full picture of the visual attention variable statistics and demographic characteristics, it is difficult to make a judgement about the reliability of the results.

- Lines: 129-131: You are listing three different measures - maintained attention, counts for fixations, and counts for visits. Then, in the next sentence, you are saying both (referring to two) measures are acceptable measures to be included in the analysis. I have no doubt that visual attention variables are relevant in such analyses. However, there are important distinctions between fixation counts (within an AOI or across AOIs) and AOI visits. One can measure fixation counts within an AOI during a single or multiple visits. The other measure, visit counts, can be used (e.g., comparing two different AOIs) without incorporating any fixation counts. Please clarify the exact measure that you are using.

- Section 2.3. By “maintained attention towards food” are you referring to fixation duration as used in Tobii Pro manuals?

- Section 3.3. Heat map grid:

o Plot C – Duration for fixations and visits. Is this the duration of the total number of fixations and total number of visits within an AOI/grid? Again, the number of fixations and number of visits are conventionally defined differently. Unless you are assuming different definitions in this particular study. Please explain.

o Plot D – Again, I am not sure how the number of fixations and visit can be measured as one variable. One can visually attend to (visit) an AOI, and fixate multiple times during that one visit. Please clarity.

o How was the irregular size of the product spaces on real store shelves mapped into the current grid system? If the space above or below the purchased product includes more than one product, how were situations like those handled. I would be interested in examining the actual heatmaps and the manual mapping of the fixations and the image of the purchased product shelves.

- Section 2.1. When was the experiment conducted? Was the recruitment done right at the store?

- Section 3.2.

o What is the measure of shelf height in Table 2?

o Is the unplanned purchase duration (in seconds) measured when taking the item from the shelf and placing in the cart?

o It would be useful to include a column that defines these variables.

o First time use of VAS in table 2. Please explain what it means.

Reviewer #3: The study evaluates the visual attention for unplanned purchases in a grocery store using mobile eye tracking technology and related the visual attention to the energy density of the food. I believe this is an interesting study. Some aspects of the study were not clear to me and are included in my comments to the authors.

ABSTRACT

• Line 9: attention bias. If attention is influenced by energy density and weight status, why is it necessarily a bias?

• Line 13: Can you explain more about the interaction: “Interaction effects between weight status and energy density for fixation and visit duration were found”

INTRODUCTION

• Line 22, can you better define unplanned purchases?

• Line 33: what about food purchases, how many are unplanned?

• Line 52: initial orientation? What do you mean?

• Line 55: “Gearhardt et al. (2012) showed a decreased initial attention towards fried food for participants with higher BMI”, decreased compared to what?

• Line 69: you hypothesize that is differs, but can you be more specific? Do you think VA will be higher or lower?

• Line 87: In what way is there in interaction? What is the direction? Can you be more specific of your H3?

METHOD

• How did you know which purchases were unplanned?

• Line 36: why no constraints? What do you mean by this?

• Line 127: what is the measurement error etc.

• Line 132: “first fixation”, do you mean “time to first fixation”?

• Line 149-150: what is the disadvantage using no overlap? What about peripheral vision? What about eye tracking errors? Should it not be accounted for a bitmore by making the AOIs a bit larger than the products themself?

RESULTS

• Was there a relation between attention and choice and if so, was this influenced by the weight status?

• Line 177: “ During these trips, 16 participants showed unplanned purchase behavior and three participants showed no unplanned purchase behavior at all” . How do you know it is unplanned? How did you measure this? How did you define it?

• Line 218 identi-fied: typo

• The product in the middle is row P. Did each isle have 7 shelves? What if a chosen product is on the bottom shelf, then there is no P-1, P-2, P-3. It is not clear how you addressed this.

• Line 222: “The product one row above”. Do you mean one shelf higher?

• Line 261: “ normal weight noticed the food of their choice earlier and looked longer at it than participants with overweight or obesity”. Due to this result, you confirm H1. However in H1 you only state that the VA would differ. Did you have a specific hypothesis how it would differ? In which case would it be higher? Does the results match this hypothesis?

• Line 235-236: “ While information labels and price tags were on average focused by three spontaneous purchase decisions (M = 3.1, SD = 5.1), the products themselves were focused more than twice as often (M = 7.3, SD = 11.9).” This sentence is not clear. The labels and price tags were focused by purchase decisions? Please rephrase.

• Check, sometimes VA and sometimes in full “Visual attention”. Please be consistent in the use of the abbreviations.

CONCLUSION

• Line 305: “Participants with normal weight made 76% of the unplanned purchases”. Again, I wonder how you defined unplanned purchases and how this was measured.

• What are the results for people with overweight? Does your study provide insight in how we can help them to attend to energy information?

• What about planned food purchases?

6. PLOS authors have the option to publish the peer review history of their article (what does this mean?). If published, this will include your full peer review and any attached files.

Reviewer #1: No

Reviewer #2: No

Reviewer #3: No

---

## [Author Response · Author response to Decision Letter 0]

7 Sep 2020

We thank the reviewers for their professional assessment. We have tried to respond to their suggestions as best we could. We sincerely hope that we have been able to clarify their questions and have contributed to the improvement of the manuscript.

Besides the changes in the manuscript you will also find an additional document with corresponding comments or rebuttals for each reviewer comment.

---

## [Decision Letter · Decision Letter 1]

4 Nov 2020

PONE-D-20-12685R1

Visual attention towards food during

unplanned purchases – A pilot study using mobile eye tracking technology

PLOS ONE

Dear Dr. Hummel,

Thank you for submitting your manuscript to PLOS ONE. After careful consideration, we feel that it has merit but does not fully meet PLOS ONE’s publication criteria as it currently stands. Therefore, we invite you to submit a revised version of the manuscript that addresses the points raised during the review process.

To give you timely response, we didn't wait for one of the reviewers' comments. Please just make your revision based on the comments from the two reviewers. 

We look forward to receiving your revised manuscript.

Kind regards,

Zhifeng Gao

Academic Editor

PLOS ONE

Reviewers' comments:

Reviewer's Responses to Questions

**Comments to the Author**

1. If the authors have adequately addressed your comments raised in a previous round of review and you feel that this manuscript is now acceptable for publication, you may indicate that here to bypass the “Comments to the Author” section, enter your conflict of interest statement in the “Confidential to Editor” section, and submit your "Accept" recommendation.

Reviewer #2: (No Response)

Reviewer #3: (No Response)

2. Is the manuscript technically sound, and do the data support the conclusions?

Reviewer #2: (No Response)

Reviewer #3: Yes

3. Has the statistical analysis been performed appropriately and rigorously? 

Reviewer #2: (No Response)

Reviewer #3: Yes

4. Have the authors made all data underlying the findings in their manuscript fully available?

Reviewer #2: (No Response)

Reviewer #3: No

5. Is the manuscript presented in an intelligible fashion and written in standard English?

Reviewer #2: (No Response)

Reviewer #3: Yes

6. Review Comments to the Author

Reviewer #2: (No Response)

Reviewer #3: The authors have has improved the manuscript. However, a few small questions still remain.

• You mention that you changed attention bias to visual attention. However, this is not adjusted everywhere (for example line 281). Please explain.

• The product in the middle is row P. Did each isle have 7 shelves? What if a chosen product is on the bottom shelf, then there is no P-1, P-2, P-3. It is not clear how you addressed this. You mention that a product at the bottom shelf, will not have P-1, P-2 etc. How did this impact the generation of Figure 1?

7. PLOS authors have the option to publish the peer review history of their article (what does this mean?). If published, this will include your full peer review and any attached files.

Reviewer #2: No

Reviewer #3: No

---

## [Author Response · Author response to Decision Letter 1]

25 Nov 2020

Comments to the reviewers

We thank the reviewers for their helpful and inspiring advices. We believe with your help we were able to im-prove the manuscript once again. 

Before we address the questions of the reviewers in the table below, we would like to point out an error in the old manuscript. In the old version of the manuscript, the name of the cut-off point for the division into high and low calorie foods was incorrectly defined. The cut off was labeled as 100kcal/100g. The correct label for the value is 260kcal/100g. The 260kcal/100g mark is the average value of the energy density of the bought prod-ucts in the current study and can roughly be seen as the transition point from high calorie food to lower energy density foods (Bechtold, 2014, S.5). We added this information in the paragraph in the manuscript. The correc-tion of the label has no effect on the calculations. The calculations have already been performed with the cor-rect values and only the label was incorrectly defined. We must apologize for this mistake and hope that the changes will meet the demands of the reviewers.

Below you will find our comments and answers to your questions.

Comments reviewer 2

Most of the comments have been addressed and the manuscript has been largely improved.

Basically, this major comment was not addressed and was not addressable at this stage because this issue should have been considered in the experimental de-sign stage. But thank you for recognizing this as one of the limitations of this study in the Conclusions sec-tion. We want to thank reviewer 2 for this positive feed-back. 

Unfortunately, there was no way to manipulate the arrangement in the supermarket and collect data in the real world. Collecting data in the real world setting also means having to make compromises. That’s why we controlled for this influence and discussed this is-sue at the end of the manuscript.

Thank you for addressing this comment by conduct-ing additional regression analysis. However, I do have some questions regarding the interpretation of the regression results.

First, what does the intercept measure? Taking Fixa-tion Counts as an example. How would you interpret the estimate of 10.19 (P<0.05)? Applying the stand-ard logic of interpretating the intercept in linear re-gression, the intercept of 10.19 essentially means that the average fixation counts of the sample is around 10 times (i.e., the fixation counts of a male shopper with zero BMI, without companion, purchas-ing a product with zero ED is about 10). The fixation counts in Figure 3 for low ED is <6 and about 4 for high ED (lower panel). So the regression results are consistent with Figure 3 in terms of statistical signifi-cance.

This estimate is much higher than the fixation counts plot in Figure 3 even taking into consideration of the negative impact of companion. With a companion re-duces almost three fixation counts. The negative im-pact of ED is statically significant but marginal as the coefficient is very small. 

My guess is that the variable BMI (exact) and energy density (exact) are not binary variables. They are the exact number of (self-reported) BMI of each partici-pant and the exact ED of each product purchased. Please correct me if I am wrong. 

Therefore, my suggestion would be why not run the regression model using BMI and ED dummies (say, high BMI=1, high ED=1). Then the intercept actually captures the average fixation counts (fixation dura-tion, visit counts, visit duration) for a male shopper with low BMI, without companion purchased low ED product. The interpretation of the intercepts as well as other coefficients in all four maintained attention models would be neat, interesting and comparable with your plots in Figure 3.

In addition, you mentioned in the revised manuscript that adding interaction terms do not improve model (Lines 376-382). With my suggestion of using dum-mies, I would like you to try the interaction terms of BMI and ED dummies again to see if the interaction term is significant or not. Combining the coefficients of BMI, ED and the interaction term of BMI and ED, will provide you the effect between and within group compaction that I mentioned. Reviewer 2 is right. The variables BMI (exact) and energy density (exact) are not binary variables. They are the exact number of (self-reported) BMI of each participant and the exact ED of each product pur-chased. We decided to use the exact numbers in-stead of dichotomized grouping variables which may already contain an inaccuracy due to the ‘artificial’ grouping.

Nevertheless, we would like to fulfill the reviewer's wish and calculate the regressions again with the di-chotomized variables for BMI and energy density.

The results for this analysis can be seen in tables 1a, 1b, 2a, and 2b. All tables are attached (outside the textbox) below.

We have calculated two models for each eye tracking measurement (FD, VD, FC, VC). The models have been calculated including variables for gender, the presence of a companion, BMI (dichotomized) and energy density (dichotomized) as fixed effects. We additionally included the interaction between BMI and energy density as a fixed effect in model 1. The indi-vidual level was included as random effect in all models.

The additional analyses are leading to two results.

1.) Even with dichotomized variables we cannot see any significant interaction effect for any eye tracking measurement. 

2.) Since the new calculations with dichotomous vari-ables did not result in a significant improvement of the regression models, we decided to keep the first mod-els with continuous variables.

One last information about the interpretation of the intercepts of the fixed effect in random effect models. The estimator of the fixed effect for the random ef-fects models is the appropriate estimate for the inter-cept of the random-effects. Below (inside the textbox) you will find the output of the estimates for the ‘indi-vidual intercepts’ for the random effects for the re-gression model calculated for FD. In one last step, one can calculate the mean of these values and will get the value for the intercept.

Table1a: regression models for fixation duration

 fixation duration (s)

 model 1 model 2 

fixed effects estimate SE Χ² (p) estimate SE Χ² (p)

intercept 2.60 0.48 17.88

(<.001) 2.55 0.51 16.41

(<.001)

BMI -0.98 0.56 2.92

(0.087) -0.45 0.49 0.00

(1.000)

energy 

density -0.64 0.25 6.29

(0.012) -0.46 0.23 3.93

(0.047)

companion (partner) -0.67 0.59 1.29

(0.256) -0.98 0.60 1.73 

(0.189)

gender -0.85 0.49 2.85

(0.091) -0.80 0.52 1.46

(0.227)

BMI x 

energy density 0.96 0.56 2.76

(0.097) - - -

random effect variance SD variance SD 

participant 0.39 0.62 - 0.48 0.69 -

residual 0.71 0.84 - 0.71 0.84 -

Table1b: regression models for visit duration

 visit duration (s)

 model 1 model 2 

fixed effects estimate SE Χ² (p) estimate SE Χ² (p)

intercept 3.02 0.59 18.25

(<.001) 2.95 0.62 16.98

(<.001)

BMI -1.17 0.70 2.68

(0.102) -0.56 0.56 0.00

(1.000)

energy 

density -0.77 0.37 4.22

(0.040) -0.54 0.33 2.57

(0.109)

companion (partner) -0.54 0.71 0.59

(0.443) -0.89 0.71 0.62

(0.430)

gender -0.79 0.60 1.70

(0.192) -0.74 0.63 0.33

(0.564)

BMI x 

energy density 1.09 0.77 1.85

(0.174) - - -

random effect variance SD variance SD 

participant 0.32 0.57 - 0.43 0.65 -

residual 1.61 1.27 - 1.61 1.27 -

 

Table2a: regression models for fixation count

 fixation count

 model 1 model 2 

fixed effects estimate SE Χ² (p) estimate SE Χ² (p)

intercept 7.82 1.79 13.82

(<.001) 7.62 1.83 12.17

(<.001)

BMI -3.23 2.10 2.27

(0.132) -.170 1.71 0.00

(1.000)

energy 

density -1.86 1.03 3.20

(0.074) -1.31 0.92 1.97

(0.160)

companion (partner) -1.14 2.17 0.28

(0.599) -1.97 2.13 0.00

(1.000)

gender -1.47 1.81 0.65

(0.419) -1.33 1.86 0.00

(1.000)

BMI x 

energy density 2.65 2.20 1.41

(0.235) - - -

random effect variance SD variance SD 

participant 4.25 2.06 - 4.67 2.18 -

residual 11.96 3.46 - 12.02 3.05 -

Table2b: regression models for visit count

 visit count

 model 1 model 2 

fixed effects estimate SE Χ² (p) estimate SE Χ² (p)

intercept 3.12 0.44 24.35

(<.001) 3.09 0.46 24.25

(<.001)

BMI -.053 0.52 0.99

(0.319) -0.22 0.42 0.41

(0.522)

energy 

density -0.52 0.28 3.40

(0.065) -0.40 0.25 2.55

(0.110)

companion (partner) -0.41 0.53 0.58

(0.447) -0.59 0.53 1.37

(0.242)

gender -0.81 0.45 2.988

(0.084) -0.77 0.46 2.97

(0.085)

BMI x 

energy density 0.55 0.57 0.85

(0.357) - - -

random effect variance SD variance SD 

participant 0.19 0.44 - 0.24 0.49 -

residual 0.88 0.94 - 0.87 0.93 -

 

Comments reviewer 3

The authors have has improved the manuscript. However, a few small questions still remain. We thank reviewer 3 for this positive feedback. 

• You mention that you changed attention bias to visual attention. However, this is not adjusted everywhere (for example line 281). Please explain. We thank the reviewer for the attentive reading and this note. Accordingly, we have revised corresponding parts in the Manuscript. See lines 281-282 and line 462-463.

• The product in the middle is row P. Did each isle have 7 shelves? What if a chosen product is on the bottom shelf, then there is no P-1, P-2, P-3. It is not clear how you addressed this. You mention that a product at the bottom shelf, will not have P-1, P-2 etc. How did this impact the generation of Figure 1? The product in the middle of the standardized heat map is always the bought product (and abbreviated as P).

Not every isle in the supermarket had 7 shelves or was 11 product rows wide. Some isles even had more than 7 shelves and many isles had more than 11 product rows. Thus, the heat maps represent a standardized and reduced section of the shelf walls in the supermarket surrounding the pur-chased product. The aspect ratio of the heat maps was chosen as land-scape format according to human vision. 

Average relative shelve height is 0.6 (see table 2). Further measures of dispersion for relative shelf height are: min=0.14, max=1, Q1=0.33, Q3=0.85. Thus, the middle 50% of all bought products have been posi-tioned in the relative shelve height between 0.33 and 0.85. (see Boxplot for relative shelve height).

The results show what we already know from former studies. The majority of the purchased products were found in the middle shelf area, slightly above the middle of the shelves (which is at 0.5). Some measurements were found in the lower and some in the upper areas. Since the values are quite equally distributed over the entire height of the shelves, the effect on the production of the heat maps should be quite small and negligible.

References

Bechtold, A. (2014). Energiedichte der Nahrung und Körpergewicht Wissenschaftliche Stellungnahme der DGE. https://www.ernaehrungs-umschau.de/fileadmin/Ernaehrungs-Umschau/pdfs/pdf_2014/01_14/EU01_2014_M014_M023_-_002d_011d.qxd.pdf

---

## [Decision Letter · Decision Letter 2]

6 Jan 2021

PONE-D-20-12685R2

Visual attention towards food during

unplanned purchases – A pilot study using mobile eye tracking technology

PLOS ONE

Dear Dr. Hummel,

Thank you for submitting your manuscript to PLOS ONE. After careful consideration, we feel that it has merit but does not fully meet PLOS ONE’s publication criteria as it currently stands. Therefore, we invite you to submit a revised version of the manuscript that addresses the points raised during the review process.

It seems that there is one issue of the experimental design that cannot be fixed. Please carefully address this issue and the related reviewer's comments. I understand the difficulty of considering all the factors when running an experiment in field, but the limitations should be clearly discussed so readers understand the potential issues. 

We look forward to receiving your revised manuscript.

Kind regards,

Zhifeng Gao

Academic Editor

PLOS ONE

Reviewers' comments:

Reviewer's Responses to Questions

**Comments to the Author**

1. If the authors have adequately addressed your comments raised in a previous round of review and you feel that this manuscript is now acceptable for publication, you may indicate that here to bypass the “Comments to the Author” section, enter your conflict of interest statement in the “Confidential to Editor” section, and submit your "Accept" recommendation.

Reviewer #2: (No Response)

Reviewer #3: All comments have been addressed

2. Is the manuscript technically sound, and do the data support the conclusions?

Reviewer #2: Partly

Reviewer #3: Yes

3. Has the statistical analysis been performed appropriately and rigorously? 

Reviewer #2: Yes

Reviewer #3: Yes

4. Have the authors made all data underlying the findings in their manuscript fully available?

Reviewer #2: Yes

Reviewer #3: Yes

5. Is the manuscript presented in an intelligible fashion and written in standard English?

Reviewer #2: Yes

Reviewer #3: Yes

6. Review Comments to the Author

Reviewer #2: 1) You did briefly discuss the issue of fixed arrangement at the end of the manuscript. I also read in your response the following: “That’s why we controlled for this influence and discussed this issue at the end of the manuscript.” However, my understanding is that you did not control the influence of fixed arrangement in your experiment. Please clarify your statement.

One way to control the influence of fixed arrangement would be through randomizing the starting point of your participants’ shopping experiences. For example, if low caloric foods were located in the front sections of the store (and high caloric foods in the back), you could instruct half of your participants to start shopping from the front. The other half of the participants would then start shopping from the back of the store. This would ensure that all of your participants didn’t start by paying attention to and purchasing low caloric foods first, thus controlling for the fixed arrangements. Does this make sense?

2) Thank you for taking into consideration of my suggestion and conducted additional analysis using dichotomous variables.

First off, I want to note that the presentation of your regression results were messy. You should preview your response file before submitting. Because the results shown in Tables 1a, 1b, 2a, and 2b are not really in a table format, it took me some time to figure out which number corresponds to what. If my understanding is correct, Model 1 in Tables 1a, 1b, 2a, 2b incorporated interaction terms and thus parallel/comparable to Table 3 in your manuscript.

By comparing Model 1 results with Table 3, I understand that using dichotomous variables does not significantly alter the results and thus all the results in Table 3 are retained (i.e., MBI, energy density as continuous variables).

However, you may already realize that your Tables 1a, 1b, 2a, 2b results are actually more consistent/comparable with Figure 3. This is because intercepts in your regression measure exactly the FD, VD, FC and VC of your base group, adding (or subtracting if negative) the coefficients of BMI, and ED provides estimates for the other group.

Figure 3 are plots based on raw data, while results in Tables 1a, 1b, 2a, 2b control for individual characteristics. Hopefully this explanation make sense to you why I wanted to see regression results based on dichotomous groups.

Reviewer #3: (No Response)

7. PLOS authors have the option to publish the peer review history of their article (what does this mean?). If published, this will include your full peer review and any attached files.

Reviewer #2: No

Reviewer #3: No

---

## [Author Response · Author response to Decision Letter 2]

14 Jan 2021

Dear Reviewers, dear Editor.

With the help of the reviewers and the editor, we have been able to steadily improve the publication to this day. We are aware of this development and the understanding of the reviewers and thank them for their support throughout the whole review process. 

We understand that the paper in its current version still has some weaknesses that need to be improved before publication. Some misunderstandings and format issues have arisen. These include the issues raised by the reviewer, which we will address now. 

1) In our last review, particularly in the response letter to the reviewers, we indicated that we controlled for the influence of the arrangement. Unfortunately, we were a little too imprecise. We did not mean that we controlled for the arrangement by randomizing the starting point as reviewer 2 points out. This would have been a very appropriate control method for testing the influence of the arrangement for an experiment. Nevertheless, our study describes much more an observational study in which we as scientists have only a very limited influence on certain variables, including the arrangement in the supermarket. We were interested in observing and evaluating spontaneous purchases in situations that were as natural as possible and not ‘artificial’. A high external validity was clearly the focus of our study. The benefit of this study is therefore the recording and analysis of realistic shopping situations and behavior in a real-world setting. Therefore, we had to forgo many experimental controls such as testing the influence of the arrangement by varying the starting point. In addition, varying the starting point would also have led to people having to walk unrealistic distances, possibly even having to walk distances twice or three times to be able to buy all the products. The time of the purchase would have changed with it and as far as time and time pressure are really connected as reviewer 2 assumed, the time pressure would have changed with it as well. Thus, varying the starting point in a realistic setting does not lead to the desired controls without significant side effects. The research team had already thought about these conditions before the study and deliberately refrained from varying this factor in any way. In our opinion, this inevitably leads to more unrealistic conditions and thus contradicts our fundamental intention. Accordingly, we followed observational studies and collected such influencing variables with our data to be able to statistically control for them as part of our analyses afterwards. And that is what we mentioned when we wrote we controlled for this variable. Since the reviewer also pointed out deficiencies in the manuscript in this respect, we added an additional part in the discussion describing this issue again in more detail and justifying our approach. Furthermore, we have extended the discussion regarding time and time pressure and a possible confounding with eye movement measurements and pointed out more clearly and again possible limitations of our results. We hope we were able to satisfy the reviewers and editors sufficiently regarding this point. We hope we were able to satisfy the reviewers and editors sufficiently about this point.

2) I am personally very embarrassed by this point, especially the formatting. The way it is presented is in no way appropriate. Unfortunately, I had copied the table into the text template and assumed that the formatting would be preserved. This was not the case. In the future I will check this several times or use figures of the tables if necessary. To provide the tables at least now in an appropriate format, I attach them to this document again.

We very much hope that the additions to the manuscript and the additional descriptions in this letter will help to better describe our approach so that we can meet the requirements of the reviewers and the editor. We thank again all reviewers and the editor for their help.

---

## [Decision Letter · Decision Letter 3]

15 Feb 2021

Visual attention towards food during

unplanned purchases – A pilot study using mobile eye tracking technology

PONE-D-20-12685R3

Dear Dr. Hummel,

We’re pleased to inform you that your manuscript has been judged scientifically suitable for publication and will be formally accepted for publication once it meets all outstanding technical requirements.

Kind regards,

Zhifeng Gao

Academic Editor

PLOS ONE

Additional Editor Comments (optional):

Reviewers' comments:

Reviewer's Responses to Questions

**Comments to the Author**

1. If the authors have adequately addressed your comments raised in a previous round of review and you feel that this manuscript is now acceptable for publication, you may indicate that here to bypass the “Comments to the Author” section, enter your conflict of interest statement in the “Confidential to Editor” section, and submit your "Accept" recommendation.

Reviewer #2: All comments have been addressed

2. Is the manuscript technically sound, and do the data support the conclusions?

Reviewer #2: Yes

3. Has the statistical analysis been performed appropriately and rigorously? 

Reviewer #2: Yes

4. Have the authors made all data underlying the findings in their manuscript fully available?

Reviewer #2: No

5. Is the manuscript presented in an intelligible fashion and written in standard English?

Reviewer #2: (No Response)

6. Review Comments to the Author

Reviewer #2: Thank you for making efforts to clarify my last comments. The responses are satisfactory, and I think the present version is acceptable.

7. PLOS authors have the option to publish the peer review history of their article (what does this mean?). If published, this will include your full peer review and any attached files.

Reviewer #2: No

---

## [Editor Report · Acceptance letter]

18 Feb 2021

PONE-D-20-12685R3 

Visual attention towards food during
unplanned purchases – A pilot study using mobile eye tracking technology 

Dear Dr. Hummel:

I'm pleased to inform you that your manuscript has been deemed suitable for publication in PLOS ONE. Congratulations! Your manuscript is now with our production department. 

Kind regards, 

on behalf of

Dr. Zhifeng Gao 

Academic Editor

PLOS ONE